# Screening of Non-Conventional Yeasts on Low-Cost Carbon Sources and Valorization of Mizithra Secondary Cheese Whey for Metabolite Production

**DOI:** 10.3390/biotech14020024

**Published:** 2025-04-01

**Authors:** Gabriel Vasilakis, Rezart Tefa, Antonios Georgoulakis, Dimitris Karayannis, Ioannis Politis, Seraphim Papanikolaou

**Affiliations:** 1Laboratory of Food Microbiology and Biotechnology, Department of Food Science and Human Nutrition, Agricultural University of Athens, 11855 Athens, Greece; vasilakis.gavriil@gmail.com (G.V.); rezarttefa98@gmail.com (R.T.); antonisgeorg1@yahoo.gr (A.G.); dimika96@icloud.com (D.K.); 2Laboratory of Animal Breeding and Husbandry, Department of Animal Science, Agricultural University of Athens, 11855 Athens, Greece; i.politis@aua.gr

**Keywords:** non-conventional yeasts, glucose, lactose, glycerol, secondary cheese whey, polysaccharides, lipids, mannitol, waste valorization, circular economy

## Abstract

The production of microbial metabolites such as (exo)polysaccharides, lipids, or mannitol through the cultivation of microorganisms on sustainable, low-cost carbon sources is of high interest within the framework of a circular economy. In the current study, two non-extensively studied, non-conventional yeast strains, namely, *Cutaneotrichosporon curvatus* NRRL YB-775 and *Papiliotrema laurentii* NRRL Y-3594, were evaluated for their capability to grow on semi-defined lactose-, glycerol-, or glucose-based substrates and produce value-added metabolites. Three different nitrogen-to-carbon ratios (i.e., 20, 80, 160 mol/mol) were tested in shake-flask batch experiments. Pretreated secondary cheese whey (SCW) was used for fed-batch bioreactor cultivation of *P. laurentii* NRRL Y-3594, under nitrogen limitation. Based on the screening results, both strains can grow on low-cost substrates, yielding high concentrations of microbial biomass (>20 g/L) under nitrogen-excess conditions, with polysaccharides comprising the predominant component (>40%, *w*/*w*, of dry biomass). Glucose- and glycerol-based cultures of *C. curvatus* promote the secretion of mannitol (13.0 g/L in the case of glucose, under nitrogen-limited conditions). The lipids (maximum 2.2 g/L) produced by both strains were rich in oleic acid (≥40%, *w*/*w*) and could potentially be utilized to produce second-generation biodiesel. SCW was nutritionally sufficient to grow *P. laurentii* strain, resulting in exopolysaccharides secretion (25.6 g/L), along with dry biomass (37.9 g/L) and lipid (4.6 g/L) production.

## 1. Introduction

The vast volume of agro-industrial residues and wastewaters has always been a challenging issue, both for the industry and society, due to the high organic load. This has compelled the scientific community to explore alternative methods for waste treatment and/or valorization. Converting nutrients from these industrial wastes into new value-added materials through biotechnological processes and consequently reducing the pollutant load aligns with the principles of a circular economy [1,2,3].

Residues deriving from food or biodiesel industries consist of streams rich in assimilable carbon sources. Examples include lactose derived from cheese manufacturing, glycerol produced during biodiesel and soap production processes, or glucose-rich hydrolysates or wastewaters obtained from various plant-processing facilities [3,4,5]. Specifically, in Greece, cheese manufacturing practices result in the production of renowned cheeses like Feta, Graviera, or Kefalotyri. According to Eurostat data, the quantity of cheese produced amounted to 251.72 thousand tons in 2025 [6]. However, this process also leads to the disposal of significant amounts of cheese whey after milk coagulation and the draining of the curd. Feta cheese is the flagship product of Greek dairy production, requiring approximately 4 kg of goat and sheep milk for every 1 kg of cheese produced, highlighting that more than 750 thousand tons of cheese whey are generated annually. The quantities at the EU level reached 52.3 million tons in 2023 [6]. The predominant methods for managing whey in Greece include thermal processing for the production of whey cheeses such as Mizithra, Anthotyros, or Manouri resulting in deproteinized, golden-green, lactose-rich secondary cheese whey (SCW). Alternative uses are as feed or as a substrate in anaerobic digesters to prevent environmental deposition due to its high organic load [7].

Glycerol is the primary by-product mainly of the biodiesel industry and also in soap manufacturing. It is also produced as a by-product during the distillation of alcoholic beverages due to its co-production during alcoholic fermentation [5]. Efforts to reduce dependence on fossil fuels have led to increased biodiesel production in recent years, utilizing various substrates such as vegetable oils (1st-generation biodiesel), by-products like animal fats, non-edible seeds, lipids derived from the bioconversion of lignocellulosic biomass via oleaginous fungi, or used cooking oils (2nd-generation biodiesel), and lipids derived from (micro)algae (3rd-generation biodiesel). Biodiesel production in 2021 reached approximately 48 billion liters, with crude glycerol output estimated at around 5 billion liters [8]. Glucose is a sugar abundantly present in fruit and vegetable processing wastewaters, in molasses, or in cellulose of lignocellulosic biomass (the most abundant biomass worldwide) [9]. According to the FAO [10], 21.6% of global waste in 2016 consisted of fruit and vegetable waste, while depending on the specific product, this percentage can rise to as high as 40–50%. Lignocellulosic biomass annual production is estimated at approximately 180 billion metric tons, and its acquisition cost is very low; however, the utilization of monomers requires appropriate physical, chemical, and/or biological pretreatment [11,12]. The large quantities of these by-products and the pollution caused by their disposal in the environment have prompted the scientific community to explore new methods for their utilization.

From the perspective of microbial biotechnology, these by-products can be used by microorganisms as nutrient sources to produce a wide range of valuable metabolites [13,14,15,16,17]. Microbial strains with the potential to produce valuable products are being studied daily for their ability to utilize such substrates and generate significant bioproducts. Non-*Saccharomyces* yeasts are widely employed in microbial and industrial biotechnology, yielding high-value microbial products such as lipids, polyols, single-cell proteins, polysaccharides, organic acids, and more [5,7,18,19,20,21,22,23,24,25]. Among the aforementioned yeasts are those belonging to the genus *Cutaneotrichosporon* (Trichosporonaceae family) which includes approximately 13 species and consists of anamorphic, non-fermentative, and urease-positive basidiomycetous yeasts. These yeasts typically form true hyphae and arthroconidia but do not reproduce asexually. About half of these species are potentially pathogenic to humans, while others have significant biotechnological potential for metabolite production. Notably, strains of the non-conventional species *Cutaneotrichosporon oleaginosus* (previously known as *Cryptococcus curvatus*) or *Cutaneotrichosporon curvatus* can utilize a broad spectrum of different renewable carbon sources, like glucose, sucrose, lactose, xylose, glycerol, arabinose, or cellobiose and produce/accumulate high levels of lipids in their biomass [26,27,28,29]. Moreover, the genus *Papiliotrema*, belonging to the Rhynchogastremaceae family, includes more than 20 species, most of which are primarily known only in their yeast forms. The species *Papiliotrema laurentii* (previously known as *Cryptococcus laurentii*) is a non-conventional, non-pathogenic, non-motile, encapsulated, dimorphic yeast found in various habitats. Its ability to assimilate substrates containing glucose, xylose, arabinose, cellobiose, mannose, galactose, rhamnose, sucrose, lactose, or galacturonic acid presents significant potential for developing bioprocesses and producing metabolites such as microbial (exo)polysaccharides, lipids, and enzymes of biotechnological interest [30,31,32]. For the production of secondary metabolites (such as lipids, exopolysaccharides, antibiotics, polyols, organic acids, etc.), it is essential to limit the trophophase—the cell proliferation stage—so that carbon flow is redirected toward secondary metabolism. Cultivation under nitrogen limitation is one of the most commonly used techniques to restrict cell proliferation and direct the microorganism toward the production of specific metabolites, depending on its genetic background and the cultivation conditions. In addition to nitrogen, the depletion of other mineral elements, such as sulfur, magnesium, phosphorus, potassium, or calcium, may also promote this metabolic shift, depending on the microbial strains being studied [33,34,35].

Building upon the previously stated information, the primary aim of the current study was to investigate the behavior and physiological response of two non-extensively studied, non-conventional yeast strains belonging to the species *Cutaneotrichosporon curvatus* and *Papiliotrema laurentii* during their screening cultures on semi-defined, low-cost, commercial substrates under different nitrogen availability conditions (C/N ratio) and low-cost carbon sources employed as substrates. Microbial growth and metabolite production were studied at three C/N molar ratios—20, 80, and 160 mol/mol—in media containing lactose, glycerol, or glucose, as main carbon sources, in flask experiments. *P. laurentii* was selected, as the only strain capable to further grow under nitrogen limitation, and subsequently was fed-batch-cultivated in laboratory-scale bioreactors with Mizithra secondary cheese whey being implicated as substrate. The aim was the production of value-added metabolites, such as polysaccharides and lipids, as well as to remediate the wastewater generated from the whey cheese manufacturing process.

## 2. Materials and Methods

### 2.1. Microorganisms

The non-conventional yeast strains used in the present study were *Cutaneotrichosporon curvatus* NRRL YB-775 and *Papiliotrema laurentii* NRRL YΒ-3594, which were kindly provided by the USDA-ARS Culture Collection (Peoria, IL, USA). The microorganisms were stored at −18 °C in a 50% (*v*/*v*) glycerol solution, while they were regenerated in YPDA medium (Yeast Extract at 10 g/L; Peptone at 10 g/L; Glucose at 10 g/L; Agar at 20 g/L) at 4 °C, before use. All microbes were incubated once per month for 24–48 h at 30 °C to maintain their viability.

### 2.2. Media

The two microorganisms were initially screened on lactose-, glycerol-, and glucose-based media, under three different carbon-to-nitrogen ratios, i.e., C/N = 20, 80, 160 mol/mol, to evaluate their ability to consume the different carbon sources and grow, as well as to produce metabolites under various nitrogen contents.

#### 2.2.1. Lactose-Based Media

The cultivation of microorganisms in the main culture medium requires their prior submerged cultivation (preculture) in a nutrient-rich medium, in this case containing 10 g/L of D-lactose (Himedia Laboratories, Pvt. LTd, Maharashtra, India), 10 g/L of yeast extract (Condalab, Madrid, Spain—nitrogen content ≈ 10%, *w*/*w*), and 10 g/L of bacteriological peptone (MC024, LAB M Ltd., Heywood, UK—nitrogen content ≈ 10%, *w*/*w*). The preculture was conducted in 250 mL (Erlenmeyer) flasks (Simax Kavalier, Praha, Czech Republic) containing 50 ± 1 mL of the medium (working volume − Vw = 20%, *v*/*v*). The flasks were sterilized in an autoclave at 121.1 °C for 20 min, aseptically inoculated using a cryovial containing the strain, and incubated in an orbital shaker (Zhicheng, Shangai, China) at 30 ± 1 °C with an agitation rate of 180 ± 5 rpm for efficient oxygen transfer. The inoculation of the main medium was carried out during the exponential growth phase of the cells.

The main culture’s semi-defined substrate contained an initial lactose concentration (S_0_) of 60 ± 2 g/L, which is close to the concentration of lactose in the SCW [7], serving as the main carbon source. Nitrogen was sourced half from yeast extract (organic source) and the remainder from ammonium sulfate (Penta Chemicals, Prague, Czech Republic—inorganic source containing ≈ 21%, *w*/*w* of N). Analytical grade minerals and salts were also added according to Papanikolaou et al. [36] and the mixture was diluted in distilled water. The concentrations of all nutrients are provided in Table 1. Nitrogen source concentrations were adjusted appropriately to achieve the desired C/N ratios (C/N = 20, 80, 160 mol/mol—see Table 1). The media were prepared, transferred to 250 mL flasks (Vw = 20%, *v*/*v*), sterilized, aseptically inoculated (using 1 mL of the preculture—2%, *v*/*v*), and incubated, at the conditions described previously. The pH was maintained within the range of 5.5–6.0 using 5 M NaOH or HCl solutions.

#### 2.2.2. Glycerol-Based Media

Glycerol was also tested as a carbon main source and, in this case, the preculture was containing 10 g/L glycerol (Glycerol BP/USP Pharm, Darmstadt, Germany), 10 g/L of yeast extract, and 10 g/L of peptone (see previously). The preculturing process was conducted as described above. The main glycerol-based cultures contained glycerol at S_0_ = 60 ± 2 g/L, and yeast extract and ammonium sulfate at concentrations to achieve the desirable C/N molar ratio, and also the rest nutrients, as presented in Table 1. The media were prepared, transferred to flasks, sterilized, aseptically inoculated, and incubated, as described previously.

#### 2.2.3. Glucose-Based Media

The two strains were additionally cultivated on glucose-based media. In this case, the preculture was containing 10 g/L of D-glucose (Carlo Erba, Milano, Italy), 10 g/L of yeast extract, and 10 g/L of peptone. The preculturing process was conducted as described above. The main glucose-based cultures were containing S_0_ = 60 ± 2 g/L glucose, yeast extract, and ammonium sulfate at appropriate concentrations to achieve the three different C/N molar ratios, and the rest nutrients as presented in Table 1. The media were prepared, transferred to flasks, sterilized, aseptically inoculated, and incubated, as described previously.

#### 2.2.4. Scale-Up, Fed-Batch Cultivation on Secondary Cheese Whey

*P. laurentii* NRRL YB-3594, as the only strain to totally assimilate lactose at any C/N ratio tested, was subsequently fed-batch-cultivated in pretreated SCW. The SCW was provided by the Laboratory of Dairy Research (Agricultural University of Athens); the cheese and whey cheese-making processes were carried out as described in the previous study of Vasilakis et al. [7] to finally obtain the SCW. The SCW pretreatment process included centrifugation at 12,000 rpm, 4 °C for 20 min (Sorvall LYNX 6000 Superspeed Centrifuge, Thermo Scientific, Waltham, MA, USA), and microfiltration of supernatant using a filter with a pore diameter of 0.22 μm (Polycap AS 36 Capsule Filter, Whatman^TM^ Cytiva, Maidstone, UK). The aim of this treatment was the removal of remaining flocculated, non-bioavailable proteins and the sterilization of the medium, avoiding any thermal sterilization process in the autoclave, which would lead to further protein flocculation and the liquid’s non-enzymatic browning (caramelization) due to the Maillard reaction. The pH value, total solids content, lactose, initial free amino nitrogen (FAN_0_), and total Kjeldahl nitrogen (TKN) concentrations of pretreated SCW were determined according to Vasilakis et al. [7]. The composition and rest traits of pretreated SCW are summarized in Table 2.

The centrifuged and microfiltered sterile SCW was transferred to a previously sterilized jacketed bioreactor (Labfors Infors HT, Bottmingen, Switzerland) of 2.0 L total volume (Vw = 1.1 L—55%, *v*/*v*, of total volume), without adding extra nutrients nor minerals and salts; 100 mL of exponentially grown lactose-based preculture was inoculated (see Section 2.2.1. Lactose-based media) was used as inoculum of 1000 mL of SCW, under aseptic conditions. The cultivation conditions in the bioreactor were as follows: incubation temperature 30 ± 1 °C, stirring 600 ± 5 rpm, and aeration 1.5 vvm, which were maintained constant throughout the cultivation. To control foam formation, Antifoam 204 solution (Sigma-Aldrich, St. Louis, MO, USA) was added to the culture medium via pumps when needed. Intermittent pulse-feeding was carried out by injecting an appropriate volume of a concentrated solution containing cheese whey-derived lactose (purity of 94.0%, along with 3.5% moisture and 2.5% impurities like salts), with a concentration of 300 g/L, when the lactose concentration was below half of its initial value, under aseptic conditions.

### 2.3. Analytical Methods

Flasks were periodically removed from the orbital shaker or samples were taken from the bioreactors and the cells were harvested using centrifugation (9000 rpm at 4 °C for 10 min in a Hettich Universal Centrifuge (Model 320-R, Merck KGaA, Darmstadt, Germany)). The supernatant underwent further analyses, as will be described below. The precipitate was washed twice with distilled water, dried to constant weight, gravimetrically determined, and expressed as dry biomass (X, g/L). Total lipids (L, g/L) were extracted from the dry biomass using chloroform/methanol organic solvent mixture and were, also gravimetrically determined after solvent evaporation. Fatty acid profile was performed through GC analysis after fatty acid methyl esters (FAMEs) derivatization. The analysis was carried out in a Fisons GC 8000 Series apparatus (Ipswich, UK), equipped with a CPWAX 52 CB column (Agilent, Santa Clara, CA, USA) and a flame ionization detector (Ipswich, UK). The individual FA content was expressed as weight percentage (g/100 g of total FA or %, *w*/*w*). The total Kjeldahl nitrogen (TKN, g/L) was determined through Kjeldahl analysis in a Kjeltek^TM^ 8100 Distillation Unit (Foss A/S, Hillerød, Denmark). Cellular polysaccharides (cPS, g/L) were determined through DNS assay after chemical hydrolysis. For more information about the aforementioned analyses, consider the publication of Vasilakis et al. [7].

The collected supernatant was further analyzed; pH value, residual sugar, mannitol, and FAN concentrations were determined based on Vasilakis et al. [7]. Specifically, for residual sugar (S_R_, g/L) or released mannitol (MAN, g/L), detections and quantifications were carried out through High Performance Liquid Chromatography in a Waters Alliance 2695 apparatus (Waters Corporation, Milford, MA, USA) equipped with RI (2414 Refractive Index) detector and the molecules were determined using standard curves. In case of carbon sources, the consumed concentration (S_CON_, g/L) was calculated as the difference between S_R_ and S_0_. Also, the consumed FAN_CON_ was the difference between the residual FAN (FAN_R_) and FAN_0_.

The cultures were performed in triplicate (two in case of bioreactor), and the mean values accompanied with the standard deviation were calculated for consumed carbon sources (S_CON_, g/L), produced biomass (X, g/L), lipids (L, g/L), cellular polysaccharides (cPS, g/L), mannitol (MAN, g/L), and for consumed free amino nitrogen (FAN_CON_, mg/L). In all individual cultures, the yield or substrate-to-biomass conversion coefficient (Y_X/S_, g total dry biomass produced per g substrate consumed), the substrate-to-lipid conversion coefficient (Y_L/S_, g/g), the substrate-to-polysaccharides conversion coefficient (Y_cPS/S_, g/g), and the substrate-to-mannitol conversion coefficient (Y_MAN/S_, g/g) were determined. Additionally, the lipid content of the biomass (K_L/X_, g lipids per g dry biomass), the polysaccharide content (K_cPS/X_, g/g), the productivity values for dry biomass (P_X_, concentration of produced dry biomass per hour, mg/L/h), lipids (P_L_, concentration of lipids per hour, mg/L/h), mannitol (P_MAN_, concentration of mannitol per hour, mg/L/h), and for cellular polysaccharides (P_cPS_, concentration of cellular polysaccharides per hour, mg/L/h) were calculated, according to Papanikolaou et al. [36]. Substrate’s consumption rate, rS = −ΔS/Δt (g/L/h) [37], was determined during the fed-batch cultivation. Each experimental point in tables and figures present the mean value and standard error of the independent determinations. Statistical analysis was carried out using the SPSS for Windows statistical package program, version 22.0.0. All means were compared using *t*-test and one-way ANOVA followed by Tukey’s post hoc test (*p* < 0.05). Data were plotted using Kaleidagraph 4.0.3.0 (Synergy Software 1988–2006) showing the mean value and standard deviation.

## 3. Results

### 3.1. Growth of C. curvatus on Lactose-Based Media Under Various Nitrogen Conditions

The non-conventional, non-extensively studied yeast *C. curvatus* NRRL YB-775 was shake-flask batch-cultured on commercial lactose-based substrates (S_0_ ≈ 60 g/L) to investigate its ability to assimilate lactose, to grow, and to produce metabolites, under various molar ratios of C/N (i.e., 20, 80, and 160 mol/mol). The results of kinetics from endpoints of all cultures are recorded in Table 3. The fatty acid composition of lipids recovered from the dry biomass at the end of all cultures are presented in Table 4.

The cultivation of the strain *C. curvatus* NRRL YB-775 at a C/N molar ratio of 20 mol/mol resulted in total lactose assimilation and 27.8 g/L dry biomass after 142 h, which was containing 12.6 g/L polysaccharides (K_cPS/X_ = 45.5%, *w*/*w* of the total dry biomass) and 1.1 g/L lipids (K_L/X_ = 3.9%, *w*/*w*). The productivity of dry biomass was 196 mg/L/h and the one of cellular polysaccharides, which were the most abundantly produced metabolites under these conditions, was approximately 89 mg/L/h at the end of the cultivation. Lipid analysis revealed that the dominant fatty acid was oleic acid (42.0%, *w*/*w*), followed by palmitic (29.5%, *w*/*w*), linoleic (17.0%, *w*/*w*), and stearic acid (11.5%, *w*/*w*). At C/N ratios of 80 mol/mol and 160 mol/mol, the microorganism exhibited very slow consumption rates once nitrogen was depleted, leading to the termination of the cultures at 142 and 164 h, respectively, after approximately 55% (S_CON_ = 33.2 g/L) and 29% (S_CON_ = 17.1 g/L) of the initial lactose concentration had been assimilated. Biomass levels reached 13.0 g/L (P_X_ = 94 mg/L/h) and 8.0 g/L (P_X_ = 48 mg/L/h) and were containing 6.9 g/L (K_cPS/X_ = 51.6%, *w*/*w* − P_cPS_ = 49 mg/L/h) and 3.1 g/L (K_cPS/X_ = 39.5%, *w*/*w* − P_cPS_ = 19 mg/L/h) polysaccharides, respectively. Lipid production was 1.4 g/L (K_L/X_ = 10.5%, *w*/*w*) at C/N = 80 mol/mol and 1.8 g/L (K_L/X_ = 22.9%, *w*/*w*) at 160 mol/mol. Lipid analysis showed that at C/N ratios of 80 and 160 mol/mol, the content of the dominant fatty acid, oleic acid, was increased (47.2% and 51.3%, *w*/*w*), while the linoleic acid was decreased (14.6% and 10.0%, *w*/*w*), respectively, compared to the respective values at C/N equal to 20 mol/mol. No statistically significant differences were observed in the contents of the other two fatty acids.

### 3.2. Growth of C. curvatus on Glycerol-Based Media Under Various Nitrogen Conditions

The strain *C. curvatus* NRRL YB-775 was also shake-flask batch-cultivated on commercial glycerol-based substrates (S_0_ ≈ 60 g/L), under the three C/N ratios. The results of kinetics from endpoints of all cultures are recorded in Table 5. The fatty acid composition of lipids at the endpoint of all cultures are presented in Table 6.

The cultivation of the strain *C. curvatus* NRRL YB-775 at a C/N molar ratio of 20 mol/mol resulted in total glycerol consumption and 31.3 g/L dry biomass after 171 h, which was containing 40.1% (*w*/*w*) polysaccharides (cPS = 12.6 g/L) and 2.9% (*w*/*w*) lipids (L = 1.1 g/L). The productivity of dry biomass was 183 mg/L/h and the one of polysaccharides 73 mg/L/h at the end of the cultivation. Based on lipid analysis, the dominant fatty acid was oleic acid (39.9%, *w*/*w*), followed by palmitic (23.4%, *w*/*w*), linoleic (18.4%, *w*/*w*), and stearic acid (18.3%, *w*/*w*). At C/N ratios of 80 mol/mol and 160 mol/mol, the strain exhibited slower consumption rates after nitrogen limitation. Glycerol was totally assimilated after 331 h in case of C/N = 80 mol/mol, while the culture at 160 mol/mol was terminated at 450 h, when approximately 76% (S_CON_ = 45.6 g/L) of the initial glycerol concentration had been assimilated. Biomass levels reached 15.3 g/L (P_X_ = 46 mg/L/h) and 9.4 g/L (P_X_ = 21 mg/L/h) and were containing 7.0 g/L (K_cPS/X_ = 45.9%, *w*/*w* − P_cPS_ = 21 mg/L/h) and 4.6 g/L (K_cPS/X_ = 49.3%, *w*/*w* − 10 mg/L/h) polysaccharides, respectively. As in the previous case of the growth on lactose, even in the media adjusted with the low initial C/N molar ratio, appreciable quantities of endopolysaccharides (K_cPS/X_ = 40.1%, *w*/*w*) were recorded, with K_cPS/X_ values increasing, when higher nitrogen limitation (higher initial C/N molar ratio) was imposed into the medium. Mannitol production was detected at 5.9 g/L (P_ΜAΝ_ = 18 mg/L/h) and 6.2 g/L (P_ΜAΝ_ = 14 mg/L/h) under C/N ratios = 80 and 160 mol/mol, respectively. Lipid production was 2.1 g/L (K_L/X_ = 13.7%, *w*/*w*) at 80 mol/mol and 1.4 g/L (K_L/X_ = 14.7%, *w*/*w*) at 160 mol/mol. Lipid analysis showed that the dominant fatty acid, oleic acid, was also increased (49.8% and 52.5%, *w*/*w*), while decreased values were observed in cases of linoleic (12.6% and 9.6%, *w*/*w*) and stearic acid (13.5% and 13.6%, *w*/*w*), at C/N ratios of 80 and 160 mol/mol, respectively, compared to the values at 20 mol/mol. No statistically significant differences were observed in the contents of palmitic acid.

### 3.3. Growth of C. curvatus on Glucose-Based Media Under Various Nitrogen Conditions

The strain *C. curvatus* NRRL YB-775 was finally shake-flask batch-cultivated on commercial glucose-based substrates (S_0_ ≈ 60 g/L), under the three C/N ratios. The kinetics of glucose and FAN assimilation, as well as of biomass and mannitol production in the case of C/N = 160 mol/mol are presented in Figure 1. The results of kinetics from endpoints of all cultures (and intermediate time point at C/N = 20 mol/mol, where mannitol was produced and subsequently assimilated) are recorded in Table 7. The fatty acid composition of lipids at the endpoint of all cultures are presented in Table 8.

The nitrogen-excess cultivation of the strain *C. curvatus* NRRL YB-775 at the glucose-based substrate resulted in 25.0 g/L dry biomass after 117 h, which was containing 45.4% (*w*/*w*) polysaccharides (cPS = 11.4 g/L) and 4.3% (*w*/*w*) lipids (L = 1.1 g/L), demonstrating, once more, the potential of the tested strain, as regards the production of polysaccharides, even under nitrogen-excess conditions. The productivity of dry biomass was 214 mg/L/h and the P_cPS_ = 97 mg/L/h at the end of the cultivation. The dominant fatty acid was oleic acid (47.3%, *w*/*w*), followed by linoleic (23.6%, *w*/*w*), palmitic (18.6%, *w*/*w*), and stearic acid (10.5%, *w*/*w*). Mannitol production (MAN = 7.7 g/L) was detected at 62 h of cultivation (P_ΜAΝ_ = 124 mg/L/h). At the same time, the glucose had been totally assimilated, thus, the microbes subsequently consumed the mannitol to produce even more biomass (final X = 25.0 g/L as mentioned above). At C/N ratios of 80 mol/mol and 160 mol/mol, the strain exhibited slightly slower consumption rates after nitrogen limitation, but glucose was totally assimilated after 129 and 185 h, respectively. Biomass levels reached 15.2 g/L (P_X_ = 118 mg/L/h) and 13.5 g/L (P_X_ = 73 mg/L/h) and were containing 6.8 g/L (K_cPS/X_ = 44.6%, *w*/*w* − P_cPS_ = 53 mg/L/h) and 6.2 g/L (K_cPS/X_ = 45.9%, *w*/*w* − P_cPS_ = 34 mg/L/h) polysaccharides, respectively. Mannitol production was 8.3 g/L (P_ΜAΝ_ = 64 mg/L/h) and 13.0 g/L (P_ΜAΝ_ = 70 mg/L/h) under C/N ratios = 80 and = 160 mol/mol, respectively. Lipid production was 1.8 g/L (K_L/X_ = 11.8%, *w*/*w*) at 80 mol/mol and 2.2 g/L (K_L/X_ = 16.3%, *w*/*w*) at 160 mol/mol. Lipid analysis showed increased values at oleic (50.6% and 52.8%, *w*/*w*), palmitic (22.1% and 24.6%, *w*/*w*) and stearic acid (13.1% and 13.2%, *w*/*w*), while the content of linoleic acid was decreased (14.2% and 9.4%, *w*/*w*) at C/N ratios of 80 and 160 mol/mol, respectively, compared to the values at 20 mol/mol.

### 3.4. Growth of P. laurentii on Lactose-Based Media Under Various Nitrogen Conditions

The non-conventional, poorly explored yeast *P. laurentii* NRRL YB-3594 was shake-flask batch-cultured on commercial lactose-based substrates (S_0_ ≈60 g/L), under various molar ratios of C/N (i.e., 20, 80, and 160 mol/mol). The results of kinetics from endpoints of all cultures are recorded in Table 9. The fatty acid composition of lipids at the endpoint of all cultures are presented in Table 10.

The cultivation of the strain *P. laurentii* NRRL YB-3594 at a C/N molar ratio of 20 mol/mol resulted in total lactose assimilation and 24.7 g/L dry biomass after 81 h, which was containing 10.1 g/L polysaccharides (K_cPS/X_ = 41.0%, *w*/*w*) and 1.0 g/L lipids (K_L/X_ = 4.2%, *w*/*w*). The productivity of dry biomass was 304 mg/L/h and the one of cPS was 125 mg/L/h at the end of the cultivation. Lipid analysis revealed that the dominant fatty acid was oleic acid (40.7%, *w*/*w*), followed by palmitic (29.0%, *w*/*w*), stearic (18.3%, *w*/*w*), and linoleic acid (12.0%, *w*/*w*). At C/N ratios of 80 mol/mol and 160 mol/mol, the microorganism exhibited slightly slower consumption rates after nitrogen limitation, but lactose was totally assimilated after 135 and 219 h, respectively. Biomass levels reached 17.8 g/L (P_X_ = 132 mg/L/h) and 13.1 g/L (P_X_ = 60 mg/L/h) and were containing polysaccharides at 7.6 g/L (K_cPS/X_ = 42.7%, *w*/*w* − P_cPS_ = 56 mg/L/h) and 5.2 g/L (K_cPS/X_ = 39.7%, *w*/*w* − P_cPS_ = 24 mg/L/h), respectively. Moreover, lipid production was 1.4 g/L (K_L/X_ = 7.9%, *w*/*w*) at C/N = 80 mol/mol and 2.2 g/L (K_L/X_ = 16.8%, *w*/*w*) at 160 mol/mol. Lipid analysis showed that at C/N ratio of 80 mol/mol, statistically significant decrease was detected only in the case of linoleic acid (10.9%, *w*/*w*), while no significant differences were observed in the contents of the rest fatty acids compared to the respective values at C/N equal to 20 mol/mol. However, at C/N ratio of 160 mol/mol, oleic and palmitic acid were slightly increased (44.0% and 32.8%, *w*/*w*), while the contents of stearic and linoleic acid were decreased (13.9% and 9.3%, *w*/*w*), respectively, compared to the values at 20 mol/mol.

### 3.5. Growth of P. laurentii on Glycerol-Based Media Under Various Nitrogen Conditions

The strain *P. laurentii* NRRL YB-3594 was also shake-flask batch-cultivated on commercial glycerol-based substrates (S_0_ ≈ 60 g/L), under the three C/N ratios. The results of kinetics from endpoints of all cultures are recorded in Table 11. The fatty acid composition of lipids at the endpoint of all cultures are presented in Table 12.

The cultivation of the strain *P. laurentii* NRRL YB-3594 under nitrogen excess resulted in total glycerol consumption and 23.5 g/L dry biomass after 340 h, which was containing 47.4% (*w*/*w*) polysaccharides (cPS = 11.1 g/L) and 5.1% (*w*/*w*) lipids (L = 1.2 g/L). The productivity of dry biomass was 69 mg/L/h and the one of cellular polysaccharides 33 mg/L/h at the end of the cultivation. Based on lipid analysis, the dominant fatty acid was oleic acid (41.6%, *w*/*w*), followed by palmitic (24.3%, *w*/*w*), stearic (17.2%, *w*/*w*), and linoleic acid (17.0%, *w*/*w*). At C/N ratios of 80 mol/mol and 160 mol/mol, the strain exhibited even slower consumption rates after nitrogen limitation. The cultivations were terminated at 440 h; at that time approximately 66% (S_CON_ = 39.7 g/L) and 52% (S_CON_ = 31.2 g/L) of the initial glycerol concentration had been assimilated. Biomass levels reached 13.9 g/L (P_X_ = 32 mg/L/h) and 8.3 g/L (P_X_ = 19 mg/L/h) and were containing 6.2 g/L (K_cPS/X_ = 44.7%, *w*/*w* − P_cPS_ = 14 mg/L/h) and 3.5 g/L (K_cPS/X_ = 41.9%, *w*/*w* − 8 mg/L/h) polysaccharides, respectively. Lipid production was 1.3 g/L (K_L/X_ = 9.4%, *w*/*w*) at 80 mol/mol and 2.1 g/L (K_L/X_ = 25.1%, *w*/*w*) at 160 mol/mol. Lipid analysis showed that only linoleic acid was slightly but statistically important increased (19.8% and 18.2%, *w*/*w*) at C/N = 80 and = 160 mol/mol, respectively, compared to the values at 20 mol/mol. No statistically significant differences were observed in the contents of the rest fatty acids.

### 3.6. Growth of P. laurentii on Glucose-Based Media Under Various Nitrogen Conditions

The strain *P. laurentii* NRRL YB-3594 was also shake-flask batch-cultivated on commercial glucose-based substrates (S_0_ ≈ 60 g/L), under the three C/N ratios. The results of kinetics from endpoints of all cultures are recorded in Table 13. The fatty acid composition of lipids at the endpoint of all cultures are presented in Table 14.

The cultivation of the strain *P. laurentii* NRRL YB-3594 under nitrogen excess resulted in total glucose consumption and 21.4 g/L dry biomass after 91 h, which, interestingly and in accordance with all previous results, contained appreciable quantities of endopolysaccharides (*viz*. 49.1%, *w*/*w* in dry biomass, cPS = 10.5 g/L). The strain accumulated restricted quantities of lipids (=5.1%, *w*/*w*, in dry biomass, L = 1.1 g/L). The productivity of dry biomass was 235 mg/L/h and the P_cPS_ was 115 mg/L/h at the end of the cultivation. Based on lipid analysis, the dominant fatty acid was oleic acid (42.2%, *w*/*w*), followed by palmitic (32.1%, *w*/*w*), linoleic (13.1%, *w*/*w*), and stearic acid (12.6%, *w*/*w*). At C/N ratios of 80 mol/mol and 160 mol/mol, the strain exhibited slightly slower consumption rates after nitrogen limitation and the glucose was totally consumed after 125 and 185 h, respectively. Biomass was 16.5 g/L (P_X_ = 132 mg/L/h) and 11.9 g/L (P_X_ = 63 mg/L/h) and were containing 7.6 g/L (K_cPS/X_ = 46.1%, *w*/*w* − P_cPS_ = 61 mg/L/h) and 4.9 g/L (K_cPS/X_ = 41.2%, *w*/*w* − 26 mg/L/h) polysaccharides, respectively. Lipid production was 1.7 g/L (K_L/X_ = 10.3%, *w*/*w*) at 80 mol/mol and 2.1 g/L (K_L/X_ = 17.6%, *w*/*w*) at 160 mol/mol. Lipid analysis showed a statistically significant increase in stearic acid (18.0% and 17.1%, *w*/*w*) and decreases at the values of palmitic (28.7% and 29.0%, *w*/*w*) and linoleic acid (10.8% and 9.1%, *w*/*w*). Oleic acid was significantly increased only under C/N of 160 mol/mol (44.8%, *w*/*w*).

### 3.7. Fed-Batch Bioreactor Cultivation on Mizithra Secondary Cheese Whey and Polysaccharides Secretion by P. laurentii

The strain *P. laurentii* NRRL YΒ-3594, as the only one of the two strains capable to assimilate lactose after the depletion of available nitrogen in the substrate, was fed-batch-cultivated in pretreated secondary cheese whey, pulse-fed with concentrated lactose solution, in a bioreactor system (Vw = 1.1 L—55%, *v*/*v*, of total volume) to further study the biosynthesis and production of secondary metabolites. The pretreatment process, which included centrifugation and microfiltration (0.22 μm) to recover insoluble solids, resulted in the processed SCW containing approximately 56 g/L lactose, 1.1 g/L TKN, and 107 mg/L FAN_0_. The results from the fed-batch culture are summarized in Table 15 the fatty acid composition of lipids recovered at the end of the culture is presented in Table 16. The kinetics of lactose and FAN assimilation, as well as the disaccharide assimilation rate (rS), are presented in Figure 2, while the microbial growth kinetics, total lactose assimilation, and production of polysaccharides and lipids are shown in Figure 3.

The fed-batch bioreactor culture of *P*. *laurentii* NRRL YΒ-3594 on secondary cheese whey as growth medium resulted in the rapid assimilation of 30.4 g/L lactose within the first 21 h. Thus, three pulse-feedings with concentrated lactose solution were followed until the end of the cultivation and the total lactose consumption was 121.2 g/L. FAN_R_ concentration practically stabilized after the first 50 h (FAN_CON_ = 86 mg/L − 21 mg/L residual FAN remained unconsumed), and from that point onward, the lactose assimilation rate (rS) decreased significantly, as depicted in Figure 2. Overall, the dry microbial biomass reached a final quantity of 37.9 g/L (P_X_ = 306 mg/L/h), containing 4.6 g/L lipids (K_L/X_ = 12.1%, *w*/*w* − P_L_ = 37 mg/L/h). Cellular polysaccharide production peaked at 88 h (cPS = 17.1 g/L − K_cPS/X_ = 49.3%, *w*/*w* − P_cPS_ = 193 mg/L/h) but then partially decreased, and 14.0 g/L (K_cPS/X_ = 36.9%, *w*/*w* − P_cPS_ = 113 mg/L/h) were detected at the end of culture, as presented in Figure 3. From that time point (88 h) onward, a continuous increase in the viscosity of the fermentation broth was observed, in parallel with the decrease in polysaccharide content. At the end of the fermentation, no residual lactose or other released mono- or disaccharides were detected in the medium. Acidic chemical hydrolysis and DNS analysis were conducted on the recovered broth and, interestingly, the presence of polysaccharidic molecules in the medium was confirmed, at a concentration of 25.6 g/L. On the other hand, lipid accumulation was enhanced. The recovered lipids were rich in oleic (36.1%, *w*/*w*) and palmitic acid (35.6%, *w*/*w*), followed by stearic (19.0%, *w*/*w*) and linoleic acid (9.4%, *w*/*w*).

## 4. Discussion

Environmental pollution caused by waste disposal is a problem of global significance, and its mitigation is deemed essential for the survival of the planet and living organisms. Reducing pollution, combined with recycling, the use of environmentally friendly raw materials, and the adoption of a circular economy, could significantly improve the already critical situation. The utilization of microorganisms as bioconverters of low-cost by-products into new value-added products, which often compete with conventional polluting products, contributes to reducing the environmental footprint. This approach leads to the production of high-value commercial products and provides industries with the opportunity to manage their waste in a beneficial manner. In the present study, poorly explored non-conventional yeast strains were investigated for their ability to assimilate commercial carbon sources and grow on semi-defined nutrient substrates. Examining microbial growth on commercial substrates is an essential step prior to applying this approach to industrial by-products. Thus, lactose, glycerol, and glucose were used because they are present in many waste streams, such as those from the agro-food sector, food processing, or biodiesel industries, etc. Two yeast strains, *C. curvatus* NRRL YB-775 and *P. laurentii* NRRL YB-3594, were selected based on an extensive review of international literature. To the best of our knowledge, these strains have been studied at least minimally or have not at all been studied from a biotechnological perspective. They were evaluated for their ability to consume commercial glucose, lactose, or glycerol, as well as for their potential to produce bioproducts under various conditions of nitrogen availability in shake-flask batch cultures.

The microorganism *C. curvatus* NRRL YB-775 can catabolize all three carbon sources, albeit at different assimilation rates depending on the carbon source and the carbon-to-nitrogen ratios. In cultures with lactose as the main carbon source, the microorganism was unable to fully assimilate the disaccharide once nitrogen was consumed. HPLC analyses, on the liquid broth containing lactose during cultivation, showed no peaks for the monomers glucose or galactose, indicating that the disaccharide is transported into the cell via lactose permease and subsequently hydrolyzed intracellularly by β-galactosidase, a physiological characteristic also described by Seiboth et al. [38]. The dry biomass contained high levels of cellular polysaccharides (≥40%, *w*/*w*) regardless of the initial C/N ratio imposed into the medium. Interestingly, significant quantities of cellular polysaccharides were produced even in the medium with high nitrogen availability. In the trial with a low initial C/N molar ratio, K_cPS/X_ values of 45.5% (*w*/*w*) were recorded. This result was somewhat unexpected and warrants further investigation. According to existing theory, as observed in the case of cellular lipids [39,40], the synthesis of cellular polysaccharides typically begins only after nitrogen depletion from the growth medium [39,40]. However, this was not the case in the present trials. The highest cPS production, which was also among the highest results across all carbon sources tested, was observed in cultures with the highest dry biomass production, at a C/N ratio of 20 mol/mol. Additionally, the cPS content exceeded 50% *w*/*w* under C/N = 80 mol/mol, representing the highest value among all experiments.

Glycerol, as the main carbon source, exhibited delayed completion due to the very slow assimilation rate of the triol, especially under low nitrogen conditions. For that reason, productivity values were significantly reduced. At C/N = 20 mol/mol, the highest dry biomass (31.3 g/L) was observed among all conditions (C/N and carbon source) tested for this strain, while nitrogen-limited conditions favored mannitol production. In glucose-based cultures, the strain completely assimilated the sugar under all three nitrogen-content conditions, with the highest consumption rate observed at C/N = 20 mol/mol. For this reason, productivity values were the highest compared to experiments using the other two carbon sources. Biomass and intracellular metabolite production levels did not reach the high values observed in corresponding experiments with lactose or glycerol, as mannitol was produced in this case regardless of the C/N molar ratio. The highest mannitol production (13 g/L) was observed under strict nitrogen limitation (C/N = 160 mol/mol). At C/N = 20 mol/mol, the microorganism was consuming mannitol after glucose depletion, indicating that, although mannitol can be produced concurrently with microbial growth, nutrient-limited conditions are necessary to achieve higher titers. Simultaneously with the consumption of mannitol, an increase in polysaccharide content was observed compared to the previous time point (at 62 h). This was expected, as nitrogen availability was low to enhance the synthesis of structural proteins, thus the carbon flow of mannitol consumption was directed to polysaccharide production. Overall, dry biomass from the experiments using glycerol and glucose contained ≥40% (*w*/*w*) cellular polysaccharides, while lipid accumulation did not exceed 20% (*w*/*w*) under any condition.

Biochemically, lipid accumulation is known to be favored under restrictive growth conditions, particularly nitrogen limitation, which is the most studied form of limitation. Although strains of the genus *Cutaneotrichosporon*, such as *C. oleaginosus* ATCC 20509 (formerly *Cryptococcus curvatus* ATCC 20509), and more broadly, members of the Trichosporonaceae family, are characterized by high oleaginicity (>20% lipids on dry biomass, *w*/*w*) [2,4,7,28,29,41,42], lipid production in this study remained at low levels. Lipid accumulation increased gradually in parallel with the increase in the C/N molar ratio, reaching a maximum of 23% *w*/*w* only at C/N = 160 mol/mol. Lipid analysis revealed that oleic acid was the predominant fatty acid (≥40%, *w*/*w*) and could serve as a raw material for biodiesel production. The analysis also revealed an increase in oleic acid content and a decrease in polyunsaturated fatty acids (PUFAs), specifically linoleic acid, as the C/N ratio increased. This pattern was consistent across all experiments regardless of the carbon source. This observation suggests a correlation between oleic acid content and storage lipids/lipid accumulation, while linoleic acid appears to be associated with cell proliferation, likely due to the presence of PUFAs in cellular and organelle membranes, primarily in the sn-2 position of phospholipids [43,44,45].

The production of mannitol is particularly noteworthy, as it is a metabolite that is rarely secreted and only by strains of related species, according to the international literature. Gallego-García et al. [46] reported that the microorganism *Cryptococcus curvatus* CL6032 produced up to 28 g/L of mannitol in semi-continuous cultures using glucose and fructose as the initial substrates, with periodic glucose additions to the medium. In a later study by the same group [46], the strain was shown to produce mannitol from various agro-industrial by-products derived from tomatoes, peppers, and watermelons, with the highest mannitol yield (35 g/L) observed in high-nitrogen cultures (C/N = 15 mol/mol) using sugars derived from watermelon residues. The selection of mannitol production/secretion in contrast to lipid production/accumulation suggests that, under nitrogen-deprived conditions, this strain redirects carbon flow towards the production of fructose-6-phosphate from glucose-6-phosphate via the enzyme phosphoglucose isomerase, followed by the production of fructose through fructose-6-phosphate phosphatase. Fructose is subsequently converted to mannitol with the simultaneous consumption of one NADPH molecule, as described by Gonçalves et al. [47] and Diamantopoulou and Papanikolaou [5]. Its secretion into the culture medium is likely due to its intracellular accumulation, which may be linked to excess ATP levels in the cell, making the reconversion of mannitol to fructose unnecessary, otherwise this reaction would result in NADPH release. Beyond its role in redox balance, polyols such as mannitol have also been associated with osmoprotective and thermoprotective functions. In this study, the temperature was maintained at levels favorable for yeast growth, while the carbon source concentration might have been high for this microorganism. Experiments under identical conditions but with a lower initial concentration of glucose or glycerol could potentially reveal evidence of osmotic stress. It is noteworthy that the microorganism did not produce mannitol during the catabolism of lactose, even though lactose contains glucose monomers and would normally follow the glycolytic pathway described above. This specific metabolic behavior of this strain requires further investigation.

In the case of *P. laurentii* NRRL YB-3594, this strain was able to extensively consume all three main carbon sources, each at a different rate, which was influenced by the availability of nitrogen in the culture medium. To be more specific, glucose and lactose were completely assimilated, while glycerol was fully consumed only under the high C/N ratio of 20 mol/mol, at a significantly slower rate compared to the cultures utilizing the rest carbon sources. The nitrogen-limited cultures were terminated before the complete assimilation of glycerol due to an extremely slow consumption rate, resulting also in very low metabolite productivity values. According to the literature, there is significant diversity in the growth behavior of different yeast species, as well as different strains of the same species, when cultivated on substrates with glycerol as the main carbon source. The mechanisms of glycerol uptake have not been extensively studied in non-*Saccharomyces* yeasts. However, the composition of the growth medium likely plays a crucial role in the yeast’s metabolic behavior and may influence key genes, such as *GUT1*, which encodes glycerol kinase. The presence of nitrogenous compounds (e.g., amino acids, yeast extract, peptone) positively impacts glycerol assimilation and the growth capacity of yeasts [48]. This observation is also confirmed by the present study, where the limitation of nitrogen sources restricted glycerol assimilation and further yeast growth. Regarding lactose-based cultures, HPLC analyses of the liquid broth revealed that, also in this case the disaccharide was hydrolyzed intracellularly, as no residual monomers were detected throughout the cultivation period.

In all cultures performed at a C/N ratio of 20 mol/mol for the three carbon sources, the microorganism successfully utilized them, yielding the highest dry biomass concentrations (>20 g/L—highest observed in lactose-based cultures) as well as polysaccharide production (>10 g/L). The highest biomass and cPS productivity values among all experiments were observed in nitrogen-rich cultures using media containing lactose. Although the polysaccharide content exceeded 45% (*w*/*w*) in some cases, a slight decline was observed in experiments at C/N = 160 mol/mol. As in previous cases with *C. curvatus* NRRL YB-775, the strain *P. laurentii* NRRL YB-3594 exhibited significant intracellular polysaccharide accumulation, regardless of nitrogen availability in the medium. This contradicts the theory suggesting that a sufficient nitrogen limitation is essential for the synthesis of endopolysaccharides in noticeable quantities within yeast cells or fungal mycelia [39,40]. Under nitrogen-deprived conditions, lipid production and accumulation were slightly promoted. A gradual increase in lipid accumulation was observed as the C/N ratio increased, but only in the glycerol-based culture at C/N = 160 mol/mol did lipid accumulation exceed 20% (*w*/*w*). Lipid production remained relatively low across the nine experiments conducted, and in all cases, oleic acid was the predominant fatty acid (≥40% *w*/*w*), followed by palmitic acid, which in some cases exceeded 30% (*w*/*w*).

In contrast to the above-mentioned, Vieira et al. [49] cultivated the isolated strain *P. laurentii* UFV-1 (*P*. *laurentii* CBS 139) in two nutrient media, using glucose or xylose as the primary carbon sources. In the glucose-based medium, maximum lipid accumulation reached 24% (*w*/*w*) of the dry biomass weight (5.3 g/L) after 96 h of batch cultivation at a C/N ratio of 48 mol/mol, with the lipids being rich in oleic acid (59.3%) and palmitic acid (29.4%). Additionally, the strain *Cryptococcus laurentii* DMKU-AmC14 accumulated lipids at 28% (*w*/*w*) of dry biomass (L = 0.9 g/L) when cultured on pure glycerol at a concentration of 70 g/L [50]. Cultivation of the strain *C. laurentii* 11 in secondary cheese whey led to lipid accumulation of 28% of the dry biomass weight (L = 1.3 g/L) [51], while a very high lipid content of 70% of the dry biomass weight (L = 5.1 g/L) was observed during cultivation of the strain *C. laurentii* UCD 68-201 in secondary cheese whey [19]. Although these studies indicate that the microorganism is oleaginous (lipid accumulation >20% *w*/*w* of dry biomass), in the present study—apart from the one case mentioned above—and in the study by Vasilakis et al. [7], which investigated the growth of *P*. *laurentii* NRRL YB-3594 and *P*. *laurentii* NRRL Y-2536 on substrates with commercial lactose or pretreated secondary cheese whey as the primary carbon source, lipid accumulation did not exceed 20% of the dry biomass weight.

*P*. *laurentii* NRRL YB-3594 was successfully fed-batch-cultivated at the bioreactor system with pretreated SCW as substrate. During the pretreatment process, steam sterilization in an autoclave was rendered unnecessary due to ultrafiltration treatment (filter pore size of 0.22 μm). The substrate was not supplemented with any additional nutrients, as SCW is sufficient in all the essential nutrients required. Furthermore, during cultivation, no chemicals (HCl or NaOH) were needed for pH regulation. These interventions significantly contribute to reducing the overall cost of the bioprocess. Secretion of polysaccharides was observed, resulting in a highly viscous supernatant at the end of fermentation, which was found to be rich in polysaccharides. In flask cultures, high viscosity was not observed, due to the lower concentration of extracellular polysaccharides (EPSs), which probably resulted from the reduced initial concentration of the carbon source. During the bioreactor cultivation, the microorganism produced cellular polysaccharides (>50%, *w*/*w* in an intermediate time point of the culture) but also secreted exopolysaccharides, thus, the K_cPS/X_ coefficient was gradually reduced. Moreover, partial degradation of cellular polysaccharides occurred simultaneously with lactose consumption and an increase in cellular lipid content, suggesting a potential interplay between the biosynthesis of cellular lipids and polysaccharides. In parallel, FAN depletion enhanced lipid accumulation, which was not high, but lipid productivity values were significantly increased.

The extracellular polysaccharides produced by *C. laurentii* strains, typically in the form of a capsule surrounding the cell, enhance cellular resistance to physical and biological stresses, such as salinity stress. These polysaccharides are of industrial interest due to their immunochemical properties and applications in pharmaceutical, cosmetic, and food industries. Additionally, they can act as wound-healing agents or even as cryoprotective molecules [30,31]. In experiments conducted with *P. laurentii* NRRL YB-3594 and NRRL Y-2536 grown on media containing commercial lactose or pretreated secondary cheese whey as the primary carbon source, polysaccharide secretion (9.4 g/L) was observed only for the latter strain when cultivated on pretreated secondary whey at a C/N ratio of approximately 60 mol/mol [7]. Similarly, Smirnou et al. [31] reported polysaccharide secretion (4.3 g/L) by *C. laurentii* DSMZ 70,766 when grown in a bioreactor using a semi-synthetic medium with sucrose (35 g/L) as the main carbon source, with the highest dry biomass production (14.3 g/L) recorded after 120 h. In their study, Pavlova et al. [30] produced 6.4 g/EPS L when shake-flask batch-cultivated *C. laurentii* AL 100 on a sucrose-based medium at a C/N = 23 mol/mol. These findings highlight the variability in metabolite accumulation and/or secretion among different strains of *P. laurentii*, which depends on their genetic background and is influenced by the cultivation conditions.

## 5. Conclusions

To sum up, this paper presents a preliminary study on the ability of two non-conventional yeasts to assimilate various carbon sources under different nitrogen availability conditions and to accordingly direct their anabolism toward the various bioproducts. The two studied strains are able to grow on low-cost substrates containing glucose, lactose, or glycerol, yielding high concentrations of microbial biomass under nitrogen-excess conditions, predominantly composed of polysaccharides. Glucose- and glycerol-based cultures of *C. curvatus* promote the secretion of mannitol, which is rare for strains of this species, while subsequent assimilation of mannitol as a secondary carbon source after the depletion of the primary one is observed. In contrast, cultivation in lactose does not favor mannitol production; lactose assimilation was notably reduced under nitrogen-depleted conditions—observations that require further investigation. The lipids produced by both strains were rich in oleic acid and could potentially be utilized for the production of second-generation biodiesel. However, their accumulation did not exceed 20% (*w*/*w*) of dry biomass in most cases, despite the implementation of nitrogen-limited cultivation conditions. In flask cultures of *P. laurentii*, a reduction in cellular polysaccharide content was observed under nitrogen-limited conditions, which is associated with their secretion into the culture medium. This observation was further confirmed by fed-batch experiments conducted in a bioreactor using SCW as the substrate. SCW was proven to be nutritionally complete for supporting the growth of the strain. The produced (exo)polysaccharides may hold significant interest, as suggested by the international literature. Further investigation will be conducted to perform structural characterization and evaluate their potential bioactive properties. For an in-depth investigation of the effects of the studied parameters on metabolic pathways and, consequently, on the production of metabolic products, molecular analyses (such as transcriptomics) are required, which could potentially be conducted in future research.

## Figures and Tables

**Figure 1 biotech-14-00024-f001:**
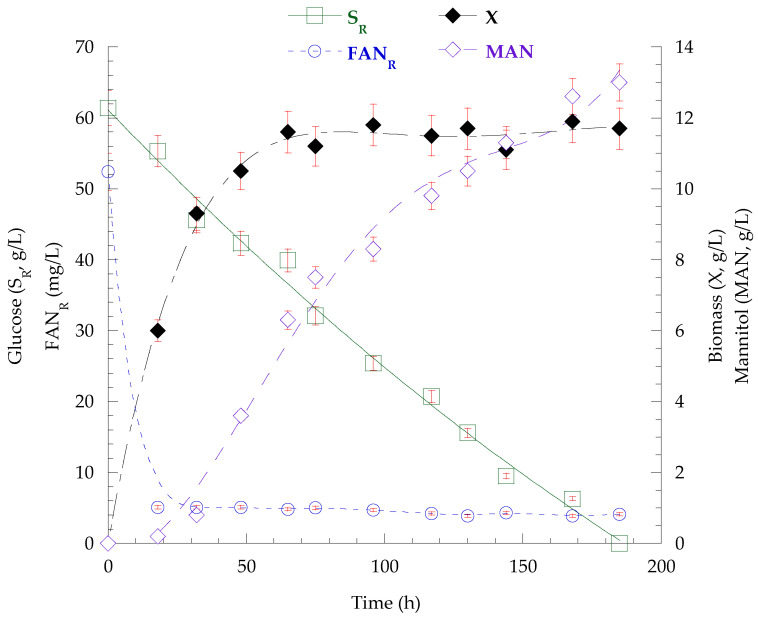
Kinetics of glucose (S_R_) and FAN_R_ concentration, as well as of the biomass and mannitol production derived from the batch cultivation of *C. curvatus* NRRL YB-775 on glucose-based medium under nitrogen limitation (C/N = 160 mol/mol).

**Figure 2 biotech-14-00024-f002:**
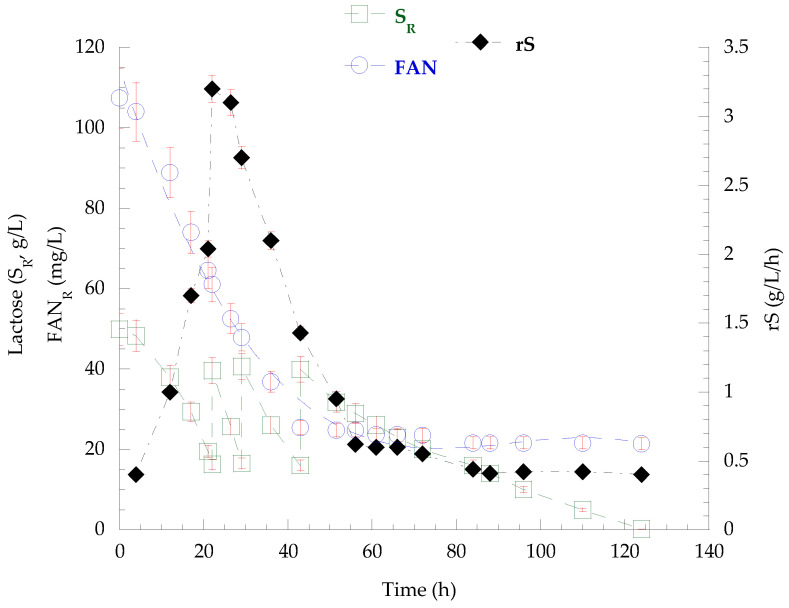
Kinetics of lactose (S_R_) and FAN_R_ concentration, as well as of the lactose consumption rate (rS) derived from the fed-batch cultivation of *P. laurentii* NRRL YB-3594 on SCW, pulse-supplemented with concentrated lactose solution.

**Figure 3 biotech-14-00024-f003:**
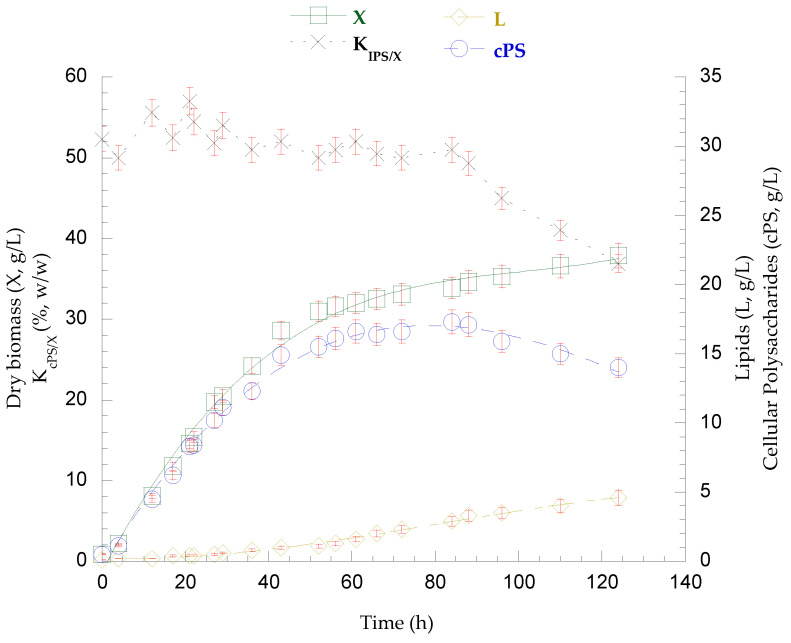
Kinetics of biomass (X), lipids (L), and cellular polysaccharide (cPS) production, as well as the polysaccharide content of the dry biomass (K_cPS/X_, %) derived from the fed-batch cultivation of *P. laurentii* NRRL YB-3594 on SCW, pulse-supplemented with concentrated lactose solution.

**Table 1 biotech-14-00024-t001:** Composition of the semi-defined culture media containing the three main carbon sources (i.e., lactose, glycerol, or glucose), the nitrogen sources (yeast extract and ammonium sulfate) for each different nitrogen content (viz. 20, 80, 160 mol/mol), and the rest minerals.

Sugar (g/L)	Lactose			Glycerol			Glucose		
	60 ± 2								
C/N ratio (mol/mol)	20	80	160	20	80	160	20	80	160
Yeast Extract (g/L)	7.4	1.8	0.9	6.8	1.7	0.9	7.0	1.7	0.9
(NH_4_)_2_SO_4_ (g/L)	3.5	0.9	0.4	3.2	0.8	0.4	3.3	0.8	0.4
KH_2_PO_4_	7.00								
Na_2_HPO_4_	2.50								
MgSO_4_ × 7H_2_O	1.50								
MnSO_4_ × H_2_O	0.06								
ZnSO_4_ × 7H_2_O	0.02								
CaCl_2_ × 2H_2_O	0.15								
FeCl_3_ × 6H_2_O	0.15								

**Table 2 biotech-14-00024-t002:** Composition and characteristics of centrifuged and filtered SCW.

Total Solids (g/L)	Lactose (g/L)	FAN_0_ (mg/L)	TKN (g/L)	pH
69.6 ± 1.2	56.5 ± 0.4	107.4 ± 3.1	1.1± 0.1	6.2 ± 0.2

**Table 3 biotech-14-00024-t003:** Quantitative data originated from the kinetics of *C. curvatus* NRRL YB-775 when batch-cultivated on semi-defined lactose-based substrates, at an initial concentration of ≈60 g/L, under three nitrogen conditions (C/N = 20, 80, 160 mol/mol). The table presents the results (mean value ± standard deviation) for each C/N molar ratio at the endpoint of the shake-flask cultures.

C/N (mol/mol)	Time (h)	S_cons_ (g/L)	X (g/L)	L (g/L)	cPS (g/L)	Y_X/S_ (g/g)	Y_L/S_ (g/g)	Y_cPS/S_ (g/g)	K_L/X_ (g/g)	K_cPS/X_ (g/g)	P_X_ (mg/L/h)	P_L_ (mg/L/h)	P_cPS_ (mg/L/h)	FAN_CON_ (mg/L)
20	142	60.4 ^a^ ± 0.1	27.8 ^a^ ± 0.3	1.1 ^a^ ± 0.1	12.6 ^a^* ± 0.2	0.46 ^a^ ± 0.01	0.02 ^a^ ± 0.00	0.21 ^a^ ± 0.00	0.039 ^a^ ± 0.001	0.455 ^a^ ± 0.010	195.8 ^a^ ± 2.1	7.7 ^a^ ± 0.8	89.1 ^a^ ± 1.0	416 ^a^ ± 16
80	142	33.2 ^b^ ± 0.6	13.4 ^b^ ± 0.2	1.4 ^b^ ± 0.0	6.9 ^b^ ± 0.1	0.40 ^a^ ± 0.02	0.04 ^b^ ± 0.00	0.21 ^ab^ ± 0.01	0.105 ^b^ ± 0.001	0.516 ^b^* ± 0.014	94.4 ^b^ ± 1.3	9.9 ^b^ ± 0.0	48.7 ^b^ ± 0.6	93 ^b^ ± 4
160	164	17.1 ^c^ ± 0.3	7.9 ^c^ ± 0.3	1.8 ^c^ ± 0.2	3.1 ^c^ ± 0.0	0.46^a^ ± 0.03	0.10 ^c^ ± 0.02	0.18 ^b^ ± 0.01	0.229 ^c^* ± 0.034	0.395 ^c^ ± 0.013	48.2 ^c^ ± 1.8	11.0 ^b^ ± 1.2	18.9 ^c^ ± 0.0	56 ^c^ ± 1

Different letters (a–c) indicate statistically significant differences according to Tukey’s test (*p* < 0.05). Asterisk (*) indicate the statistically highest values among all results obtained from the two tested conditions (carbon-to-nitrogen ratio and carbon source) for each strain individually.

**Table 4 biotech-14-00024-t004:** Fatty acid composition of the cellular lipids produced by *C*. *curvatus* NRRL YB-775, when batch-cultivated on lactose-containing media, under three nitrogen conditions (C/N = 20, 80, 160 mol/mol) in shake-flask experiments.

C/N (mol/mol)	Time (h)	g/100 g of Total FA					
		C16:0	C18:0	^Δ9^C18:1	^Δ9,12^C18:2	SFA	UFA
20	142	29.5 ^a^ ± 1.4	11.5 ^a^ ± 0.3	42.0 ^a^ ± 1.4	17.0 ^a^ ± 0.7	41.0 ^a^ ± 1.8	59.0 ^a^ ± 2.0
80	142	27.3 ^a^ ± 1.1	10.9 ^a^ ± 0.3	47.2 ^b^ ± 1.0	14.6 ^b^ ± 0.2	39.1 ^a^ ± 1.5	60.9 ^a^ ± 1.2
160	164	27.4 ^a^ ± 1.2	11.3 ^a^ ± 0.1	51.3 ^c^ ± 1.4	10.0 ^c^ ± 0.1	31.5 ^b^ ± 1.3	68.5 ^b^ ± 1.5

Different letters (a–c) indicate statistically significant differences according to Tukey’s test (*p* < 0.05).

**Table 5 biotech-14-00024-t005:** Quantitative data originated from the kinetics of *C. curvatus* NRRL YB-775, when batch-cultivated on semi-defined glycerol-based substrates, at an initial concentration of ≈60 g/L, under three nitrogen conditions (C/N = 20, 80, 160 mol/mol). The table presents the results (mean value ± standard deviation) for each C/N molar ratio at the endpoint of the shake-flask cultures.

C/N (mol/mol)	Time (h)	S_cons_ (g/L)	X (g/L)	L (g/L)	cPS (g/L)	Man (g/L)	Y_X/S_ (g/g)	Y_L/S_ (g/g)	Y_cPS/S_ (g/g)	Y_Man/S_ (g/g)	K_L/X_ (g/g)	K_cPS/X_ (g/g)	P_X_ (mg/L/h)	P_cPS_ (mg/L/h)	P_Man_ (mg/L/h)	FAN_CON_ (mg/L)
20	171	58.2 ^a^ ± 0.1	31.3 ^a^* ± 0.4	0.9 ^a^ ± 0.1	12.6 ^a^* ± 0.3	0.0 ^a^ ± 0.0	0.54 ^a^* ± 0.01	0.02 ^a^ ± 0.00	0.22 ^a^ ± 0.01	0.00 ^a^ ± 0.00	0.029 ^a^ ± 0.003	0.401 ^a^ ± 0.016	183.0 ^a^ ± 2.3	73.4 ^a^ ± 2.0	0.0 ^a^ ± 0.0	438 ^a^ ± 16
80	331	59.1 ^a^ ± 0.3	15.3 0 ^b^ ± 0.2	2.1 ^b^ ± 0.1	7.0 ^b^ ± 0.1	5.9 ^b^ ± 0.3	0.26 ^b^ ± 0.00	0.04 ^b^ ± 0.00	0.12 ^b^ ± 0.00	0.10 ^b^ ± 0.01	0.137 ^b^ ± 0.009	0.459 ^b^ ± 0.011	46.2 ^b^ ± 0.6	21.2 ^b^ ± 0.3	17.8 ^b^ ± 0.9	95 ^b^ ± 4
160	450	45.6 ^b^ ± 0.4	9.4 ^c^ ± 0.1	1.4 ^c^ ± 0.0	4.6 ^c^ ± 0.0	6.2 ^b^ ± 0.2	0.21 ^c^ ± 0.00	0.03 ^ab^ ± 0.00	0.10 ^b^ ± 0.00	0.14 ^c^ ± 0.00	0.147 ^b^ ± 0.004	0.493 ^c^ ± 0.002	21.0 ^c^ ± 0.1	10.3 ^c^ ± 0.0	13.9 ^c^ ± 0.3	59 ^c^ ± 1

Different letters (a–c) indicate statistically significant differences according to Tukey’s test (*p* < 0.05). Asterisk (*) indicate the statistically highest values among all results obtained from the two tested conditions (carbon-to-nitrogen ratio and carbon source) for each strain individually.

**Table 6 biotech-14-00024-t006:** Fatty acid composition of the cellular lipids produced by *C*. *curvatus* NRRL YB-775, when batch-cultivated on glycerol-containing media, under three nitrogen conditions (C/N = 20, 80, 160 mol/mol) in shake-flask experiments.

C/N (mol/mol)	Time (h)	g/100 g of Total FA					
		C16:0	C18:0	^Δ9^C18:1	^Δ9,12^C18:2	SFA	UFA
20	171	23.4 ^a^ ± 1.0	18.3 ^a^ ± 1.2	39.9 ^a^ ± 1.6	18.4 ^a^ ± 1.1	41.7 ^a^ ± 2.2	58.3 ^a^ ± 2.8
80	331	24.1 ^a^ ± 0.9	13.5 ^b^ ± 0.3	49.8 ^b^ ± 1.0	12.6 ^b^ ± 0.2	37.6 ^ab^ ± 1.3	62.4 ^ab^ ± 1.2
160	450	23.2 ^a^ ± 0.4	13.6 ^b^ ± 0.1	52.5 ^b^ ± 1.2	9.6 ^c^ ± 0.1	36.8 ^b^ ± 0.5	63.2 ^b^ ± 1.4

Different letters (a–c) indicate statistically significant differences according to Tukey’s test (*p* < 0.05).

**Table 7 biotech-14-00024-t007:** Quantitative data originated from the kinetics of *C. curvatus* NRRL YB-775, when batch-cultivated on semi-defined glucose-based substrates, at an initial concentration of ≈60 g/L, under three nitrogen conditions (C/N = 20, 80, 160 mol/mol). The table presents the results for each C/N molar ratio at the endpoint of the shake-flask cultures, as well as at an intermediate time point in case of C/N equal to 20 mol/mol, where mannitol production and subsequent assimilation was observed.

C/N (mol/mol)		Time (h)	S_cons_ (g/L)	X (g/L)	L (g/L)	cPS (g/L)	Man (g/L)	Y_X/S_ (g/g)	Y_L/S_ (g/g)	Y_cPS/S_ (g/g)	Y_Man/S_ (g/g)	K_L/X_ (g/g)	K_cPS/X_ (g/g)	P_X_ (mg/L/h)	P_cPS_ (mg/L/h)	P_Man_ (mg/L/h)	FAN_CON_ (mg/L)
20	i	62	61.3 ± 0.2	21.7 ± 0.2	0.9 ± 0.0	7.5 ± 0.1	7.7 ± 0.2	0.35 ± 0.01	0.02 ± 0.00	0.12 ± 0.01	0.13 ± 0.00	0.042 ± 0.001	0.346 ± 0.008	350.1 ± 3.1	121.0 ± 1.6	124.2 ± 3.2	316 ± 4
	ii	117	61.3 ^a^ ± 0.2	25.0 ^a^ ± 0.1	1.1 ^a^ ± 0.0	11.4 ^a^ ± 0.2	0.0 ^a^ ± 0.0	0.41 ^a^ ± 0.00	0.02 ^a^ ± 0.00	0.19 ^a^ ± 0.01	0.00 ^a^ ± 0.00	0.043 ^a^ ± 0.001	0.454 ^a^ ± 0.009	213.7 ^a^* ± 0.8	97.3 ^a^* ± 1.9	0.0 ^a^ ± 0.0	373 ^a^ ± 11
80		129	61.9 ^a^ ± 0.3	15.2 ^b^ ± 0.2	1.8 ^b^ ± 0.1	6.8 ^b^ ± 0.1	8.3 ^b^ ± 0.2	0.25 ^b^ ± 0.00	0.03 ^ab^ ± 0.00	0.11 ^b^ ± 0.00	0.13 ^b^ ± 0.01	0.118 ^b^ ± 0.008	0.446 ^a^ ± 0.014	117.8 ^b^ ± 1.6	52.7 ^b^ ± 0.8	64.3 ^b^ ± 1.6	83 ^b^ ± 7
160		185	61.4 ^a^ ± 0.2	13.5 ^c^ ± 0.2	2.2 ^b^ ± 0.2	6.2 ^b^ ± 0.2	13.0 ^c^* ± 0.4	0.22 ^b^ ± 0.01	0.04 ^b^ ± 0.01	0.10 ^b^ ± 0.01	0.21 ^c^ ± 0.02	0.163 ^c^ ± 0.017	0.459 ^a^ ± 0.022	73.0 ^c^ ± 1.1	33.5 ^c^ ± 1.1	70.3 ^b^* ± 2.1	49 ^c^ ± 3

Different letters (a–c) indicate statistically significant differences according to Tukey’s test (*p* < 0.05). Asterisk (*) indicate the statistically highest values among all results obtained from the two tested conditions (carbon-to-nitrogen ratio and carbon source) for each strain individually.

**Table 8 biotech-14-00024-t008:** Fatty acid composition of the cellular lipids produced by *C*. *curvatus* NRRL YB-775, when batch-cultivated on glucose-containing media, under three nitrogen conditions (C/N = 20, 80, 160 mol/mol) in shake-flask experiments.

C/N (mol/mol)	Time (h)	g/100 g of Total FA					
		C16:0	C18:0	^Δ9^C18:1	^Δ9,12^C18:2	SFA	UFA
20	117	18.6 ^a^ ± 0.4	10.5 ^a^ ± 1.1	47.3 ^a^ ± 1.2	23.6 ^a^ ± 0.1	29.1 ^a^ ± 1.4	70.9 ^a^ ± 1.3
80	129	22.1 ^b^ ± 0.9	13.1 ^b^ ± 0.5	50.6 ^ab^ ± 0.6	14.2 ^b^ ± 0.4	35.2 ^b^ ± 1.5	64.8 ^b^ ± 1.1
160	185	24.6 ^c^ ± 0.3	13.2 ^b^ ± 0.2	52.8 ^b^ ± 1.1	9.4 ^c^ ± 0.1	37.8 ^b^ ± 0.5	62.2 ^b^ ± 1.2

Different letters (a–c) indicate statistically significant differences according to Tukey’s test (*p* < 0.05).

**Table 9 biotech-14-00024-t009:** Quantitative data originated from the kinetics of *P. laurentii* NRRL YB-3594, when batch-cultivated on semi-defined lactose-based substrates, at an initial concentration of ≈60 g/L, under three nitrogen conditions (C/N = 20, 80, 160 mol/mol). The table presents the results for each C/N molar ratio at the endpoint of the shake-flask cultures.

C/N (mol/mol)	Time (h)	S_cons_ (g/L)	X (g/L)	L (g/L)	cPS (g/L)	Y_X/S_ (g/g)	Y_L/S_ (g/g)	Y_cPS/S_ (g/g)	K_L/X_ (g/g)	K_cPS/X_ (g/g)	P_X_ (mg/L/h)	P_L_ (mg/L/h)	P_cPS_ (mg/L/h)	FAN_CON_ (mg/L)
20	81 ^a^	58.7 ± 0.2	24.7 ^a^* ± 0.2	1.0 ^a^ ± 0.1	10.1 ^a^ ± 0.2	0.42 ^a^ ± 0.01	0.02 ^a^ ± 0.00	0.17 ^a^ ± 0.00	0.042 ^a^ ± 0.003	0.410 ^a^ ± 0.010	304.4 ^a^* ± 3.0	12.7 ^a^ ± 0.9	124.7 ^a^* ± 2.5	374 ^a^ ± 11
80	135 ^b^	59.4 0.3	17.8 ^b^ ± 0.3	1.4 ^a^ ± 0.1	7.6 ^b^ ± 0.2	0.30 ^b^ ± 0.01	0.02 ^ab^ ± 0.01	0.13 ^b^ ± 0.01	0.079 ^b^ ± 0.007	0.427 ^a^ ± 0.019	131.9 ^b^ ± 2.2	10.4 ^a^ ± 0.7	56.3 ^b^ ± 1.5	89 ^b^ ± 6
160	219 ^c^	58.9 ± 0.1	13.1 ^c^ ± 0.2	2.2 ^b^ ± 0.2	5.2 ^c^ ± 0.1	0.22 ^c^ ± 0.01	0.04 ^b^ ± 0.01	0.09 ^c^ ± 0.00	0.168 ^c^ ± 0.018	0.397 ^a^ ± 0.014	59.8 ^c^ ± 0.9	10.1 ^a^ ± 0.9	23.7 ^c^ ± 0.4	48 ^c^ ± 5

Different letters (a–c) indicate statistically significant differences according to Tukey’s test (*p* < 0.05). Asterisk (*) indicate the statistically highest values among all results obtained from the two tested conditions (carbon-to-nitrogen ratio and carbon source) for each strain individually.

**Table 10 biotech-14-00024-t010:** Fatty acid composition of the cellular lipids produced by *P. laurentii* NRRL YB-3594, when batch-cultivated on lactose-containing media, under three nitrogen conditions (C/N = 20, 80, 160 mol/mol) in shake-flask experiments.

C/N (mol/mol)	Time (h)	g/100 g of Total FA					
		C16:0	C18:0	^Δ9^C18:1	^Δ9,12^C18:2	SFA	UFA
20	81	29.0 ^a^ ± 0.9	18.3 ^a^ ± 0.7	40.7 ^a^ ± 0.6	12.0 ^a^ ± 0.2	47.3 ^a^ ± 1.6	52.7 ^a^ ± 0.7
80	135	29.6 ^a^ ± 0.7	18.1 ^a^ ± 1.0	41.4 ^ab^ ± 1.0	10.9 ^b^ ± 0.4	47.7 ^a^ ± 1.7	52.3 ^a^ ± 2.4
160	219	32.8 ^b^ ± 0.7	13.9 ^b^ ± 1.0	44.0 ^b^ ± 1.2	9.3 ^c^ ± 0.4	46.7 ^a^ ± 1.6	53.3 ^a^ ± 1.7

Different letters (a–c) indicate statistically significant differences according to Tukey’s test (*p* < 0.05).

**Table 11 biotech-14-00024-t011:** Quantitative data originated from the kinetics of *P. laurentii* NRRL YB-3594, when batch-cultivated on semi-defined glycerol-based substrates, at an initial concentration of ≈60 g/L, under three nitrogen conditions (C/N = 20, 80, 160 mol/mol). The table presents the results for each C/N molar ratio at the endpoint of the shake-flask cultures.

C/N (mol/mol)	Time (h)	S_cons_ (g/L)	X (g/L)	L (g/L)	cPS (g/L)	Y_X/S_ (g/g)	Y_L/S_ (g/g)	Y_cPS/S_ (g/g)	K_L/X_ (g/g)	K_cPS/X_ (g/g)	P_X_ (mg/L/h)	P_L_ (mg/L/h)	P_cPS_ (mg/L/h)	FAN_CON_ (mg/L)
20	340	59.4 ^a^ ± 0.2	23.5 ^a^ ± 0.4	1.2 ^a^ ± 0.1	11.1 ^a^ ± 0.2	0.40 ^a^ ± 0.00	0.02 ^a^ ± 0.00	0.19 ^a^ ± 0.00	0.051 ^a^ ± 0.005	0.474 ^a^ ± 0.015	69.1 ^a^ ± 1.2	3.5 ^a^ ± 0.3	32.8 ^a^ ± 0.4	344 ^a^ ± 20
80	440	39.7 ^b^ ± 0.6	13.9 ^b^ ± 0.5	1.3 ^a^ ± 0.1	6.2 ^b^ ± 0.2	0.35 ^b^ ± 0.02	0.03 ^a^ ± 0.01	0.1 ^b^ ± 0.00	0.094 ^b^ ± 0.010	0.447 ^ab^ ± 0.031	31.5 ^b^ ± 1.2	3.0 ^a^ ± 0.2	14.1 ^b^ ± 0.5	91 ^b^ ± 7
160	440	31.2 ^c^ ± 0.4	8.3 ^c^ ± 0.2	2.1 ^b^ ± 0.1	3.5 ^c^ ± 0.1	0.27 ^c^ ± 0.01	0.07 ^b^ ± 0.00	0.11 ^c^ ± 0.01	0.251 ^c^* ± 0.021	0.419 ^b^ ± 0.025	19.0 ^c^ ± 0.3	4.8 ^b^ ± 0.2	8.0 ^c^ ± 0.2	50 ^c^ ± 4

Different letters (a–c) indicate statistically significant differences according to Tukey’s test (*p* < 0.05).

**Table 12 biotech-14-00024-t012:** Fatty acid composition of the cellular lipids produced by *P. laurentii* NRRL YB-3594, when batch-cultivated on glycerol-containing media, under three nitrogen conditions (C/N = 20, 80, 160 mol/mol) in shake-flask experiments.

C/N (mol/mol)	Time (h)	g/100 g of Total FA					
		C16:0	C18:0	^Δ9^C18:1	^Δ9,12^C18:2	SFA	UFA
20	340	24.3 ^a^ ± 0.9	17.2 ^a^ ± 0.7	41.6 ^a^ ± 1.1	17.0 ^a^ ± 0.2	41.5 ^a^ ± 1.6	58.5 ^a^ ± 1.4
80	440	23.6 ^a^ ± 0.7	16.6 ^a^ ± 0.5	40.0 ^a^ ± 1.0	19.8 ^b^ ± 0.8	40.2 ^a^ ± 1.3	59.8 ^a^ ± 1.8
160	440	25.0 ^a^ ± 1.1	16.1 ^a^ ± 0.6	40.7 ^a^ ± 0.5	18.2 ^ab^ ± 0.4	41.1 ^a^ ± 1.6	58.9 ^a^ ± 0.9

Different letters (a,b) indicate statistically significant differences according to Tukey’s test (*p* < 0.05).

**Table 13 biotech-14-00024-t013:** Quantitative data originated from the kinetics of *P. laurentii* NRRL YB-3594, when batch-cultivated on semi-defined glucose-based substrates, at an initial concentration of ≈60 g/L, under three nitrogen conditions (C/N = 20, 80, 160 mol/mol). The table presents the results for each C/N molar ratio at the endpoint of the shake-flask cultures.

C/N (mol/mol)	Time (h)	S_cons_ (g/L)	X (g/L)	L (g/L)	cPS (g/L)	Y_X/S_ (g/g)	Y_L/S_ (g/g)	Y_cPS/S_ (g/g)	K_L/X_ (g/g)	K_cPS/X_ (g/g)	P_X_ (mg/L/h)	P_L_ (mg/L/h)	P_cPS_ (mg/L/h)	FAN_CON_ (mg/L)
20	91 ^a^	61.4 ± 0.2	21.4 ^a^ ± 0.4	1.1 ^a^ ± 0.0	10.5 ^a^ ± 0.3	0.35 ^a^ ± 0.01	0.02 ^a^ ± 0.00	0.17 ^a^ ± 0.01	0.051 ^a^ ± 0.000	0.491 ^a^ ± 0.014	235.2 ^a^ ± 4.4	12.1 ^a^ ± 0.0	115.4 ^a^ ± 3.3	326 ^a^ ± 13
80	125 ^b^	59.0 ± 0.1	16.5 ^b^ ± 0.3	1.7 ^b^ ± 0.2	7.6 ^b^ ± 0.1	0.28 ^b^ ± 0.01	0.03 ^ab^ ± 0.00	0.13 ^b^ ± 0.00	0.103 ^b^ ± 0.014	0.461 ^a^ ± 0.021	132.0 ^b^ ± 2.4	13.6 ^a^ ± 1.6	60.8 ^b^ ± 0.8	85 ^b^ ± 3
160	188 ^c^	58.7 ± 0.3	11.9 ^c^ ± 0.1	2.1 ^b^ ± 0.0	4.9 ^c^ ± 0.1	0.20 ^c^ ± 0.01	0.04 ^b^ ± 0.00	0.08 ^c^ ± 0.01	0.176 ^c^ ± 0.002	0.412 ^b^ ± 0.013	63.3 ^c^ ± 0.5	11.2 ^a^ ± 0.0	26.1 ^c^ ± 0.5	50 ^c^ ± 4

Different letters (a–c) indicate statistically significant differences according to Tukey’s test (*p* < 0.05).

**Table 14 biotech-14-00024-t014:** Fatty acid composition of the cellular lipids produced by *P. laurentii* NRRL YB-3594, when batch-cultivated on glucose-containing media, under three nitrogen conditions (C/N = 20, 80, 160 mol/mol) in shake-flask experiments.

C/N (mol/mol)	Time (h)	g/100 g of Total FA					
		C16:0	C18:0	^Δ9^C18:1	^Δ9,12^C18:2	SFA	UFA
20	91	32.1 ^a^ ± 1.1	12.6 ^a^ ± 0.5	42.2 ^a^ ± 1.7	13.1 ^a^ ± 0.1	44.7 ^a^ ± 1.7	55.3 ^a^ ± 1.8
80	125	28.7 ^b^ ± 1.1	18.0 ^b^ ± 0.7	42.5 ^a^ ± 0.5	10.8 ^b^ ± 0.4	46.7 ^a^ ± 1.8	53.3 ^a^ ± 1.0
160	188	29.0 ^b^ ± 0.6	17.1 ^b^ ± 0.3	44.8 ^a^ ± 1.0	9.1 ^c^ ± 0.6	46.1 ^a^ ± 0.8	53.9 ^a^ ± 1.6

Different letters (a–c) indicate statistically significant differences according to Tukey’s test (*p* < 0.05).

**Table 15 biotech-14-00024-t015:** Quantitative data originated from the kinetics of *P. laurentii* NRRL YB-3594, when fed-batch-cultivated on secondary cheese whey, pulse-supplemented with concentrated lactose solution. The table presents results from selective successive time points of the bioreactor cultures.

Time (h)	pH	S_cons_ (g/L)	X (g/L)	L (g/L)	cPS (g/L)	Y_X/S_ (g/g)	Y_L/S_ (g/g)	Y_cPS/S_ (g/g)	K_L/X_ (g/g)	K_cPS/X_ (g/g)	P_X_ (mg/L/h)	P_L_ (mg/L/h)	P_cPS_ (mg/L/h)	FAN_CON_ (mg/L)
21	6.1 ^a^	30.4 ^a^ ± 1.1	14.6 ^a^ ± 0.5	0.4 ^a^ ± 0.2	8.3 ^a^ ± 0.3	0.4 8 ^a^ ± 0.05	0.01 ^a^ ± 0.01	0.27 ^a^ ± 0.02	0.032 ^a^ ± 0.011	0.570 ^a^ ± 0.040	696.2 ^a^ ± 22.8	20.9 ^a^ ± 7.6	396.8 ^a^ ± 12.7	49 ^a^ ± 4
56	6.5 ^b^	92.4 ^b^ ± 0.9	31.6 ^b^ ± 0.4	1.3 ^b^ ± 0.1	16.1 ^b^ ± 0.2	0.34 ^b^ ± 0.01	0.01 ^a^ ± 0.01	0.17 ^b^ ± 0.01	0.040 ^a^ ± 0.005	0.510 ^a^ ± 0.012	564.3 ^b^ ± 7.1	22.6 ^a^ ± 2.4	287.8 ^b^ ± 3.3	82 ^b^ ± 7
88	7.3 ^c^	107.4 ^c^ ± 1.3	34.6 ^c^ ± 0.6	3.3 ^c^ ± 0.2	17.1 ^b^ ± 0.4	0.32 ^b^ ± 0.01	0.03 ^b^ ± 0.00	0.16 ^b^ ± 0.01	0.095 ^b^ ± 0.008	0.493 ^a^ ± 0.022	393.2 ^c^ ± 6.8	37.2 ^b^ ± 2.6	193.8 ^c^ ± 5.1	86 ^b^ ± 2
124	7.3 ^c^	121.2 ^d^ ± 0.1	37.9 ^d^ ± 0.5	4.6 ^d^ ± 0.2	14.0 ^c^ ± 0.3	0.31 ^b^ ± 0.01	0.04 ^b^ ± 0.00	0.12 ^c^ ± 0.00	0.121 ^c^ ± 0.002	0.369 ^b^ ± 0.013	305.9 ^d^ ± 3.8	36.7 ^b^ ± 2.0	112.9 ^d^ ± 2.4	86 ^b^ ± 3

Different letters (a–d) indicate statistically significant differences between different time points of the culture according to Tukey’s test (*p* < 0.05).

**Table 16 biotech-14-00024-t016:** Fatty acid composition of the cellular lipids produced by *P. laurentii* NRRL YB-3594, when fed-batch-cultivated on secondary cheese whey, pulse-supplemented with concentrated lactose solution.

Time (h)	g/100 g of Total FA					
	C16:0	C18:0	^Δ9^C18:1	^Δ9,12^C18:2	SFA	UFA
188	35.6 ± 1.4	19.0 ± 0.8	36.1 ± 1.0	9.4 ± 0.3	54.6 ± 2.3	45.5 ± 1.3

## Data Availability

The original contributions presented in this study are included in the article. Further inquiries can be directed to the corresponding author.

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
