# Peer review of "Screening of Non-Conventional Yeasts on Low-Cost Carbon Sources and Valorization of Mizithra Secondary Cheese Whey for Metabolite Production"

_biotech, 2025, doi:10.3390/biotech14020024_

Round 1
Reviewer 1 Report
Comments and Suggestions for Authors
This manuscript describes the production of various metabolites by non-conventional yeast strains using glucose-, glycerol-, and lactose-based media, as well as low-cost carbon sources (MSCW). The manuscript is well-structured, with experimental data effectively supporting the results and conclusions. However, to further enhance the quality of the manuscript, the following issues should be addressed:
Introduction:
While this study aims to evaluate the impact of carbon-to-nitrogen (C/N) molar ratios on metabolite production, specifically polysaccharides and lipids, in non-conventional yeasts, the introduction lacks critical background on why C/N ratios significantly influence these biosynthetic pathways. The introduction should establish how C/N ratios affect cellular metabolism and why optimizing these ratios is essential for enhancing the production of target compounds in non-conventional yeasts.
Materials and methods:
1) In Section 2.2, the authors reference three microorganisms (line 128), which contradicts the experimental design that utilized only two yeast strains. Please recheck and correct.
2) Section 2.2.4 describes scale-up production in a 2-L bioreactor with stirring at 600 rpm. However, the manuscript fails to address foam formation during cultivation—a common challenge in yeast fermentation at this scale and agitation rate. Please include details on whether foam occurred and specify the antifoam strategy employed (if possible).
3) The manuscript requires comprehensive gas chromatography (GC) methodology for fatty acid analysis. Please add a detailed subsection specifying the GC conditions, column used, and so on.
4) All data tables should include appropriate statistical analysis. Please incorporate statistical analysis results in the Table.
Results:
1) The statement on Page 6, lines 274-275 that 'No statistically significant differences were observed...' is unsupported, as no statistical analysis methodology or results are presented anywhere in the manuscript. This critical omission undermines the validity of all comparative claims. Please conduct appropriate statistical analyses (e.g., ANOVA with post-hoc tests or t-tests depending on your experimental design) and incorporate the results into all tables.
2) The statement on Page 8, line 311, and Page 12, lines 419-420 are also the same as mentioned above. Please conduct appropriate statistical analyses (e.g., ANOVA with post-hoc tests or t-tests depending on your experimental design) and incorporate the results into all tables.
3) Scientific names of microorganisms should be written in italics throughout the manuscript. Additionally, after the first full mention of a genus name, it should be abbreviated in subsequent references. Please review the entire manuscript to ensure consistency with these conventions.
4) Consider adding a summary table at the end of the results section to highlight the key findings of this study, which would enhance clarity and accessibility for readers.
Author Response
Dear Reviewer,
We would like to thank you for your time and comments, which we will attempt to address promptly.
Introduction:
While this study aims to evaluate the impact of carbon-to-nitrogen (C/N) molar ratios on metabolite production, specifically polysaccharides and lipids, in non-conventional yeasts, the introduction lacks critical background on why C/N ratios significantly influence these biosynthetic pathways. The introduction should establish how C/N ratios affect cellular metabolism and why optimizing these ratios is essential for enhancing the production of target compounds in non-conventional yeasts.
Response: As you have correctly observed, the metabolic behavior of the two non-conventional yeast strains was studied under different nitrogen conditions. The ‘Introduction’ section has been expanded to include details on how nitrogen availability (through the various low to high C/N molar ratios) influence cellular metabolism and why selecting specific ratios is crucial for enhancing the production of target compounds (L106-114).
Materials and methods:
1) In Section 2.2, the authors reference three microorganisms (line 128), which contradicts the experimental design that utilized only two yeast strains. Please recheck and correct.
Response: You are right, two strains were screened. The error has been corrected (L138).
2) Section 2.2.4 describes scale-up production in a 2-L bioreactor with stirring at 600 rpm. However, the manuscript fails to address foam formation during cultivation—a common challenge in yeast fermentation at this scale and agitation rate. Please include details on whether foam occurred and specify the antifoam strategy employed (if possible).
Response: As you correctly mentioned, cultivation in pretreated Secondary Cheese Whey at such an agitation speed (600 rpm) and aeration induces foaming in the liquid, as commonly observed in most bioreactor cultures under aerobic conditions. The bioreactor system we used is equipped with pumps and appropriate sensors to automatically add antifoam to the culture medium whenever foam levels become excessive. The text has been revised as recommended (L208-210).
3) The manuscript requires comprehensive gas chromatography (GC) methodology for fatty acid analysis. Please add a detailed subsection specifying the GC conditions, column used, and so on.
Response: As noted at the end of the paragraph (L232-233), further details on each procedure can be found in the paper by Vasilakis et al., 2022 (10.3390/app122211471). In that study, all procedures are described in detail; therefore, we aimed to avoid overloading this paper with redundant information. The text has been slightly enriched as recommended (L226-228).
4) All data tables should include appropriate statistical analysis. Please incorporate statistical analysis results in the Table.
Response: You are absolutely right. Appropriate statistical analysis was conducted using SPSS software, and different letters (a–d) were used to indicate statistically significant differences according to Tukey’s test (p < 0.05) (L259-262 & Tables’ footnotes).
Results:
1) The statement on Page 6, lines 274-275 that 'No statistically significant differences were observed...' is unsupported, as no statistical analysis methodology or results are presented anywhere in the manuscript. This critical omission undermines the validity of all comparative claims. Please conduct appropriate statistical analyses (e.g., ANOVA with post-hoc tests or t-tests depending on your experimental design) and incorporate the results into all tables.
Response: The data have been analyzed properly, as recommended.
2) The statement on Page 8, line 311, and Page 12, lines 419-420 are also the same as mentioned above. Please conduct appropriate statistical analyses (e.g., ANOVA with post-hoc tests or t-tests depending on your experimental design) and incorporate the results into all tables.
Response: The data have been analyzed properly, as recommended.
3) Scientific names of microorganisms should be written in italics throughout the manuscript. Additionally, after the first full mention of a genus name, it should be abbreviated in subsequent references. Please review the entire manuscript to ensure consistency with these conventions.
Response: You are right, we have properly corrected any unintentional error that occurred during the transfer of the text into the journal's manuscript format. The genus’ names have been abbreviated in subsequent references, as recommended.
4) Consider adding a summary table at the end of the results section to highlight the key findings of this study, which would enhance clarity and accessibility for readers.
Response: Your suggestion is valid; however, we believe that the key highlights are already presented in the Abstract, key contributions, and Conclusions. Additionally, the most important results are thoroughly analyzed in the Discussion section. Moreover, the way the results are presented in the Tables allows readers to easily refer to any experiment and, consequently, the corresponding Table to find the results of interest, regardless of whether they are considered highlights or not. Therefore, we believe that an extra Table addition would unnecessarily increase the length of an already extensive paper, thus, asterisks (*) have been added in specific Tables’ data to indicate the statistically highest values among all results obtained from the two tested conditions (carbon-to-nitrogen ratio and carbon source) for each strain individually.
Reviewer 2 Report
Comments and Suggestions for Authors
This paper studied the growth and metabolite production capacity of two non-traditional yeast strains (Cutaneotrichosporon curvatus NRRL YB-775 and Papiliotrema laurentii NRRL Y-3594) on low-cost carbon sources. In particular, the use of cheese secondary whey (SCW) for bioconversion provides a new idea for bioconversion of industrial waste. some questions and doubts remain,
1, The two yeast strains (C. curvatus and P. laurentii) are presented together without clear justification for their combined study.
2, Their metabolic pathways (mannitol vs. polysaccharide/lipid production) and industrial applications are distinct but not rigorously separated in the Results/Discussion.
3, Data (e.g., biomass yield, metabolite concentrations) lack measures of variability (e.g., standard deviation, error bars) or statistical significance (e.g., p-values).No description of statistical methods (e.g., ANOVA, t-tests) in the Materials & Methods section.
4, Abbreviations (e.g., SCW) are confusing in use
Author Response
Dear Reviewer,
We would like to thank you for your time and comments, which we will attempt to address promptly.
1, The two yeast strains (C. curvatus and P. laurentii) are presented together without clear justification for their combined study.
Response: This study, as you have correctly mentioned, focuses on the investigation of non-conventional (non-Saccharomyces) and poorly studied yeast strains that could be utilized in microbial and industrial biotechnology applications for the production of high-value microbial products. Both strains meet these criteria, which is why they were selected. We have attempted to clarify this further in the text (L88-89).
2, Their metabolic pathways (mannitol vs. polysaccharide/lipid production) and industrial applications are distinct but not rigorously separated in the Results/Discussion.
Response: This paper presents a preliminary study on the ability of these yeast strains to assimilate various carbon sources under different nitrogen availability conditions and to accordingly direct their anabolism toward the aforementioned bioproducts. This study yielded numerous observations, which we have attempted to discuss as thoroughly as possible in the Discussion section. For an in-depth investigation of the effects of the studied parameters on metabolic pathways and, consequently, on the production of metabolic products, molecular analyses (such as transcriptomics) are required, which could potentially be conducted in future research. The aforementioned has been stated at the ‘Conclusions’ section (L722-724, L743-746)
3, Data (e.g., biomass yield, metabolite concentrations) lack measures of variability (e.g., standard deviation, error bars) or statistical significance (e.g., p-values).No description of statistical methods (e.g., ANOVA, t-tests) in the Materials & Methods section.
Response: You are absolutely right. Appropriate statistical analysis was conducted using SPSS software, and different letters (a–d) were used to indicate statistically significant differences according to Tukey’s test (p < 0.05) (L259-262 & Tables’ footnotes).
4, Abbreviations (e.g., SCW) are confusing in use
Response: Abbreviations, such as SCW or those related to coefficients (YX/S, KL/X,PX, etc.), among others, are used to avoid repetitive use of full terms throughout the text, thereby enhancing clarity and preventing unnecessary redundancy that could make the reading experience cumbersome. To facilitate their understanding, an ‘Abbreviations’ section has been added in accordance with the journal's guidelines (L894).
Reviewer 3 Report
Comments and Suggestions for Authors
Attached below some comments.

Author Response
Dear Reviewer,
We would like to thank you for your time, your kind words and the comments, which we will attempt to address promptly.
- There are many abbreviations throughout the paper. I suggest adding a dedicated section to summarize them for clarity.
Response: An ‘Abbreviations’ section has been added, as recommended, based on journal's guidelines (L894).
- I suggest enriching the discussion on yeast metabolism by referencing the following works:
https://doi.org/10.1016/j.indcrop.2020.113030
https://doi.org/10.4014/jmb.1808.08015
Response: The manuscript has been enriched, as recommended .
- Some minor comments below.
Line 112: Why is this sentence in italics?
Response: It was an unintentional error that occurred during the transfer of the text into the journal's manuscript format. It has been corrected properly (L123).
Line 128-129: Please, specify the media origin.
Response: All information regarding the concentrations of the components and their origin is detailed in Sections 2.2.1–2.2.3 and presented in Table 1.
Lines 276, 312, 389: Please ensure that the names of the microorganisms are written in italics
Response: You are right, they were unintentional errors that have been corrected properly.
Round 2
Reviewer 2 Report
Comments and Suggestions for Authors
it is ok